# Automatic Segmentation of Head and Neck Tumors and Nodal Metastases in PET-CT scans

**Vincent Andrearczyk**[1] **Valentin Oreiller**[1,2] **Martin Vallières**[3,4] **Joel Castelli**[5,6,7] **Hesham Elhalawani**[8] **Mario Jreige**[2] **Sarah Boughdad**[2] **John O. Prior**[2] **Adrien Depeursinge**[1,2]

[1] *University of Applied Sciences Western Switzerland (HES-SO), Sierre, Switzerland*
[2] *Centre Hospitalier Universitaire Vaudois (CHUV), Lausanne, Switzerland*
[3] *Medical Physics Unit, McGill University, Montréal (Qc), Canada*
[4] *Département d'informatique, Université de Sherbrooke, Sherbrooke (Qc), Canada*
[5] *Radiotherapy Department,Cancer Institute Eugène Marquis, Rennes, France*
[6] *INSERM, U1099, Rennes, France*
[7] *University of Rennes 1, LTSI, Rennes, France*
[8] *Cleveland Clinic Foundation, Department of Radiation Oncology, Cleveland, OH, USA*

## Abstract

Radiomics, the prediction of disease characteristics using quantitative image biomarkers from medical images, relies on expensive manual annotations of Regions of Interest (ROI) to focus the analysis. In this paper, we propose an automatic segmentation of Head and Neck (H&N) tumors and nodal metastases from FDG-PET and CT images. A fully-convolutional network (2D and 3D V-Net) is trained on PET-CT images using ground truth ROIs that were manually delineated by radiation oncologists for 202 patients. The results show the complementarity of the two modalities with a statistically significant improvement from 48.7% and 58.2% Dice Score Coefficients (DSC) with CT- and PET-only segmentation respectively, to 60.6% with a bimodal late fusion approach. We also note that, on this task, a 2D implementation slightly outperforms a similar 3D design (60.6% vs 59.7% for the best results respectively).

**Keywords:** Head and Neck cancer, Deep learning, multimodal, PET-CT, 3D segmentation

## 1. Introduction

Head and Neck (H&N) cancers are among the most common cancers worldwide ($5^{th}$ leading cancer by incidence) (Parkin et al., 2005). Radiotherapy combined with cetuximab has been established as standard treatment (Bonner et al., 2010). However, locoregional failures remain a major challenge and occur in up to 40% of patients in the first two years after the treatment (Chajon et al., 2013). Radiomics is the quantitative extraction of high-dimensional mineable data from medical images (Lambin et al., 2012; Vallières et al., 2017). Recently, several radiomics studies based on FluoroDeoxyGlucose (FDG) Positron Emission Tomography (PET) and Computed Tomography (CT) imaging were proposed to better identify patients with a worse prognosis in a non-invasive fashion and by reusing images acquired for diagnosis and treatment planning (Vallières et al., 2017; Bogowicz et al., 2017; Castelli et al., 2017). Although highly promising, these methods were validated on 100-400 patients. Further validation on larger cohorts (e.g. 300-3000 patients) is required to

respect an adequate ratio between the number of variables and observations in order to avoid an overestimation of the generalization performance. Achieving such a validation requires the manual delineation of primary tumors and nodal metastases for every patient in three dimensions, which is intractable and error-prone. Methods for automated lesion segmentation in medical images have been proposed in various contexts, often achieving expert-level performance (Heimann and Meinzer, 2009; Menze et al., 2014). Surprisingly few studies evaluated the performance of computerized automated segmentation of tumor lesions in PET and CT images, where only a dozen approaches reported its feasibility, e.g. (Song et al., 2013; Blanc-Durand et al., 2018; Moe et al., 2019). In this paper, we propose an automatic volumetric segmentation of both tumor and nodal metastasis in PET-CT images. By focusing on metabolic and morphological tissue properties respectively, FDG-PET and CT modalities include complementary and synergistic information for cancerous lesion segmentation. This motivates the use of a bimodal fully convolutional network (V-Net).

In previous work, automated PET-CT analysis has been proposed for different tasks, including lung cancer segmentation in (Kumar et al., 2019; Li et al., 2019; Zhao et al., 2018; Zhong et al., 2018) and bone lesion detection in (Xu et al., 2018). A thorough review of deep learning for multi-modal medical image segmentation, including PET-CT tumor segmentation can be found in (Zhou et al., 2019). In (Moe et al., 2019), a PET-CT segmentation was proposed for a task similar to the one presented in this paper, i.e. H&N Gross Tumour Volume (GTV) delineation of the primary tumor as well as metastatic lymph nodes using a 2D U-Net architecture. We expand this work by comparing, on a publicly available dataset, 2D and 3D recent segmentation architectures (V-Net) as well as the complementarity of the two modalities with quantitative and qualitative analyses. Finally, we evaluate the generalization of the trained algorithms to new centers.

## 2. Methods

In this section, we describe the dataset and algorithm developed for the 3D automatic segmentation of the H&N tumors.

### 2.1. Dataset

The dataset was proposed in (Vallières et al., 2017) in the context of a radiomics study and is available on The Cancer Imaging Archive (TCIA)[1] (Clark et al., 2013; Vallieres et al., 2017). The radiotherapy contours are used for ground truth in our segmentation training and evaluation. They were drawn by expert radiation oncologists, either directly on the CT of the PET-CT study (31% of the patients) or on a different CT scan dedicated to treatment planning (69%). In the latter scenario, the contours were registered to the PET-CT scans (more details in (Vallières et al., 2017)). Initially, one PET-CT study, as well as a dedicated radiotherapy planning CT, were available for 300 patients from four institutions in Québec with histologically proven H&N cancer. In our study, we focus on oropharynx tumors for a total of 202 patients. The distribution of cases per center is 72, 57, 54 and 19. The motivation to focus on the oropharyngeal region is to operate in a controlled anatomical

---

1. https://tinyurl.com/yc5sq5jy, as of May 2020

context. We use a leave-one-center-out cross-validation to estimate the generalization to unseen centers (see Section 2.4).

## 2.2. Pre-processing

The CT and PET volumes are resampled to an isotropic $1 \times 1 \times 1$mm voxel spacing using trilinear interpolation. The CT volumes are clipped in the range $[-150, 150]$ Hounsfield Units (HU). Each image is then cropped to a volume of size $144 \times 144 \times 144$ voxels containing all tumors (primary tumor and metastatic lymph nodes), and includes the oropharyngeal region by centering the volumes around a minimal bounding box containing these tumors. A single ground truth binary mask is obtained per sample as the union of GTVs and metastatic lymph nodes and further resampled similarly to the PET and CT images to an isotropic volume. All CT and PET volumes are standardized as proposed in (Nyúl et al., 2000), i.e. for a given split, an averaged histogram mapping is learned from the training volumes and used to map the voxels of all training and test volumes to a standardized version.

## 2.3. Network Architecture

We use NiftyNet (Gibson et al., 2018) to implement a 3D V-Net (Milletari et al., 2016) and a 2D version (referred to as 2D V-Net as it has a similar architecture as the 3D V-Net for fair comparison). V-Net is a 3D volumetric fully-convolutional CNN based on and improved from its 2D counterpart (U-Net)(Ronneberger et al., 2015). The architecture is composed of four downsampling blocks (compression), four upsampling blocks (decompression) and one final prediction residual convolutional block. Downsampling and upsampling are performed by convolution layers (respectively convolution and transposed convolution with $2 \times 2 \times 2$ filters and a stride of 2) for a total of 30 convolution layers with ReLU activations and final softmax activation. In our experiments, the input and output sizes are $144 \times 144 \times 144$ ($144 \times 144 \times 1$ for the 2D version). We either use a bimodal network with both PET and CT as multiple input channels (referred to as early fusion) or a single modality as input. We also evaluate a late fusion prediction by averaging the voxel-wise probability outputs of two distinct PET and CT networks.

## 2.4. Training Scheme

We use a standard training scheme and hyper-parameters as described in the following. The loss used for computing the gradients is a combination of binary Dice Similarity Coefficient (DSC) and Cross-Entropy (CE) losses defined in (Isensee et al., 2018) and computed as $\mathcal{L}_{DSC} + \mathcal{L}_{CE}$. This dual loss benefits from both the smooth and bounded gradients of the cross-entropy loss and the explicit optimization of the Dice score used for evaluation and its robustness to class imbalance. The cross-entropy loss is computed as

$$\mathcal{L}_{CE} = \sum_i y_i \log \hat{y}_i + (1 - y_i) \log(1 - \hat{y}_i), \tag{1}$$

where $\hat{y} \in [0, 1]$ is the softmax output, $y \in \{0, 1\}$ is the ground truth mask and the sum is computed over all voxels. The Dice loss is computed as

$$\mathcal{L}_{DSC} = -2 \frac{\sum_i \hat{y}_i y_i}{\sum_i \hat{y}_i + \sum_i y_i}. \tag{2}$$

We train the networks with an Adam optimizer (Kingma and Ba, 2014) with a batch size of 12, a learning rate of 0.0003 for 200 iterations, which corresponds to the training loss plateau. The number of iterations and batch size are increased to 1000 and 36 for the 2D implementation to account for the larger number of slices (single slice per iteration in the 2D case) as compared to full volumes (full volume per iteration in 3D). Using an NVIDIA Tesla v100 32GB and the mentioned setup and hyperparameters, the training times of the 2D and 3D multimodal V-Net (early fusion) are of 4:47 and 0:49 hours respectively. Similarly, the inference times for 19 case (i.e. one of the four centers) are of 17 and 16.8 seconds.

## 2.5. Evaluation

To evaluate the performance of the models, we use a leave-one-center-out cross-validation, i.e. for each center, we take the center as the test set and the remaining three centers as the training set. We report the DSC averaged across all patients for a repetition of 10 cross-center-validation runs. We compute the cross-center-validation to evaluate the generalization of the segmentation algorithms to unknown centers, weighted by the number of cases per center. The 95% Confidence Intervals (CI) are reported considering all individual cases, regardless of the centers and are computed on averaged scores across the 10 runs. Note that variation across runs comes from networks' initializations and orders of observations in the training batches. Similarly, a statistical analysis is performed using a paired t-test on averaged DSC scores of the 202 cases.

To mitigate some of the limitations of the dataset related to the annotations (mentioned in Section 4), we evaluate the segmentation results when low HU are removed from both the ground truth masks (i.e. re-segmentation see (Zwanenburg et al., 2020)) and the predictions, i.e. we threshold them at -150 HUs, as for the CT lower bound clipping.

## 3. Experimental Results

### 3.1. Quantitative Results

The performance results are reported in Table 1. Networks trained on a single modality (PET or CT) are compared with bimodal approaches (early or late fusion). We also compare to 2D and 3D implementations. While the resampling on the $z$ axis (perpendicular to the axial plane) is not necessary for the 2D analysis, we observe an improvement of the results as it certainly acts as a type of data-augmentation.

Table 1: Comparison of single modality and bimodal network performance of 2D and 3D V-Nets. The average DSCs, sensitivity and specificity (%) are reported with 95% CIs.

| model | modalit. | DSC | precision | recall |
|---|---|---|---|---|
| 2D/3D | CT | 48.7% $_{\pm 2.2}$/49.2% $_{\pm 2.2}$ | 52.7% $_{\pm 2.9}$/48.6% $_{\pm 2.6}$ | 54.1% $_{\pm 2.8}$/65.0% $_{\pm 2.6}$ |
| 2D/3D | PET | 58.2% $_{\pm 2.3}$/58.6% $_{\pm 2.5}$ | 59.7% $_{\pm 3.0}$/59.1% $_{\pm 2.9}$/ | 66.7% $_{\pm 2.9}$/70.2% $_{\pm 3.1}$ |
| 2D/3D | early fusion | 58.5% $_{\pm 2.2}$/58.9% $_{\pm 2.3}$ | 58.1% $_{\pm 2.9}$/59.0% $_{\pm 2.9}$/ | 70.2% $_{\pm 2.7}$/**70.8%** $_{\pm 2.8}$ |
| 2D/3D | late fusion | **60.6%** $_{\pm 2.1}$/59.7% $_{\pm 2.2}$ | **69.4%** $_{\pm 2.6}$/62.8% $_{\pm 2.8}$/ | 62.1% $_{\pm 2.9}$/69.1% $_{\pm 2.8}$ |

The results show that PET-based segmentation is largely better than the one based on CT, and the best results are obtained with a late fusion of the two modalities. The late fusion methods (both 2D and 3D V-Nets) have significantly higher DSC scores than the corresponding PET alone and CT alone methods with $p$-values of a paired t-test of $p < 0.05$. The recall is higher for the 3D architectures than its 2D counterparts (69.1% vs 62.1% for the late fusion), while the precision is lower (62.8% vs 69.4%), reflecting larger 3D predictions. This is confirmed by the corresponding average predicted volumes of interests of 1.4% and 1.1% of the total volumes. Favouring over- or under-segmentation is task dependent and can also be controlled by the decision threshold on the output probabilities. To further compare the performance of the 2D and 3D architectures, we evaluate them on subsets of tumors based on the size of the latter, namely below and above 4cm of diameter, which correspond to the limit between T2 and T3 TNM stages (Huang and O'Sullivan, 2017). We notice that the 2D approach performs particularly better than the 3D counterpart on small tumors (+16% DSC, +15% precision and +3% recall), while the two methods perform equivalently in terms of DSC on larger tumors.

The average 2D DSC scores per center are 61.9%, 59.8%, 63.3% and 51.0% (corresponding number of cases: 72, 57, 54 and 19). Based on visual inspection, the last center (least represented) particularly requires a quality control of the ground truth annotations and more training data similar to this center may be needed for better generalization.

The histogram of DSC scores obtained for all test cases by the late fusion 2D V-Net is displayed in Figure 1, showing a DSC mode around 70% and several DSC scores close to zero that make the average DSC drop to 60.6%. Our internal visual analysis of low DSC outliers motivates a curation of the dataset for future analysis (see Figure 4 and Section 4).

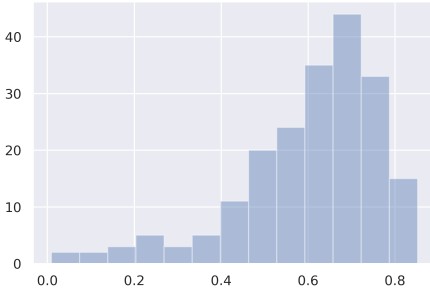

Figure 1: Count histogram of DSCs obtained by late 2D PET-CT fusion for the 4-center cross-validation.

In Figure 2, we vary the weights of the PET and CT predictions in the late fusion to investigate the respective relevance of the two modalities. For every voxel, the prediction is computed as $p_{\text{pred}} = \alpha \cdot p_{\text{CT}} + (1 - \alpha) \cdot p_{\text{PET}}$, where $p_{\text{CT}}$ and $p_{\text{PET}}$ are the probabilities of tumor prediction. Note that the results are reported in Table 1 as the mean of prediction probabilities ($\alpha = 0.5$) since we do not want to optimize hyperparameters based on test

results. Thus, Figure 2 is only reported to explore and illustrate the complementarity of the two modalities, with the best performance obtained for $\alpha = 0.4$ as well as a large drop of accuracy when giving more importance to the CT prediction.

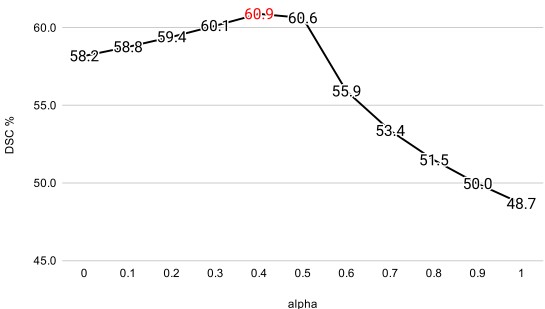

Figure 2: DSC scores for various values of weights in the PET-CT late fusion $p_{\mathrm{pred}} = \alpha \cdot p_{\mathrm{CT}} + (1 - \alpha) \cdot p_{\mathrm{PET}}$.

### 3.2. Qualitative Results

In Figure 3, we show examples of segmentation results, compared with ground truth annotations, to illustrate the benefit of combining CT and PET modalities. Figures 3 (a-c) illustrate an example of false-positive detected in the PET modality, corrected by the inclusion of CT information in early fusion. Similarly, Figures 3 (d-f) illustrate an example of large over-segmentation in the CT modality, correctly rectified by using the PET.

In Figure 4, we propose two other examples of late fusion segmentation results overlayed with a combination of PET and CT modalities (PET in *hot* colors). Figure 4 (a) shows a PET-induced false-positive example, while Figure 4 (b) shows an example of inaccurate ground truth, where the trachea is part of the tumor segmentation. The algorithm learned to discard (from other training samples) such a region that is not part of the tumor and the prediction on this example seems to segment the tumor better than the ground truth annotation. This also motivates the evaluation with thresholded masks of ground truth and predictions as reported in the quantitative results.

### 4. Discussions and Conclusion

The results show a promising potential for automatic H&N tumor and metastatic node segmentation in PET-CT images for radiomics studies and could potentially also be used for treatment planning (e.g. radiotherapy). Best results were obtained using a bimodal (late fusion of PET and CT) 2D V-Net approach with an average DSC of 60.6%. By focusing on metabolic and morphological tissue properties respectively, PET and CT modalities include complementary and synergistic information for cancerous lesion segmentation. This complementarity was highlighted by the quantitative results, where DSC scores of 48.7%,

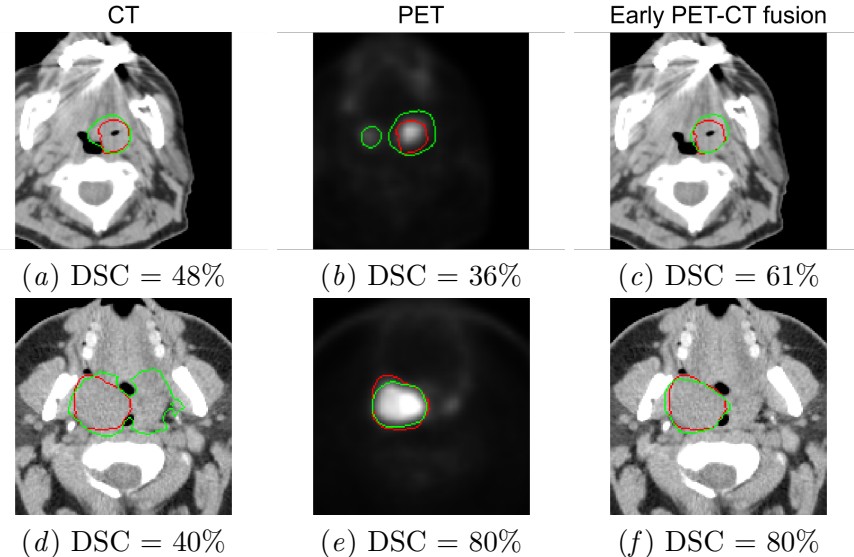

$(a)$ DSC = 48%  $(b)$ DSC = 36%  $(c)$ DSC = 61%

$(d)$ DSC = 40%  $(e)$ DSC = 80%  $(f)$ DSC = 80%

Figure 3: 2D axial examples of qualitative segmentation results (green) and ground truth (red) with their corresponding DSC (evaluated on the entire 3D volumes). Top row illustrates an example in which better prediction is obtained on the CT. Bottom row illustrates an example in which better prediction is obtained on the PET. (a,d) CT segmentation, (b,e) PET segmentation, (c,f) early fusion PET/CT segmentation.

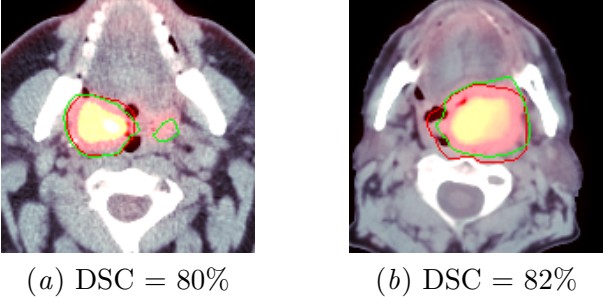

$(a)$ DSC = 80%  $(b)$ DSC = 82%

Figure 4: 2D axial examples of ground truth (red contours) and prediction (green contours) drawn on top of an overlay of CT and PET scans. (a) PET-induced false-positive segmentation and (b) accurate segmentation, with the automatic prediction discarding the trachea whereas the ground truth does not.

58.2% and 60.6% were obtained using the CT, PET and PET-CT modalities, respectively ($p$-value= 3.2e−39 and 0.000052 for a paired t-test between PET-CT and the individual modalities respectively), as well as qualitative results in Figure 3. No significant difference was obtained between the 2D and 3D approaches. One difficulty of using a 3D approach is the low resolution on the $z$-axis that we upsampled to obtain inputs with isotropic voxel dimensions (original average pixel spacing of all CT scans on x-, y- and z-axes respectively 1.09, 1.09, 2.75 mm).

Note that in (Gudi et al., 2017), inter-observer consensus DSCs of 57% and 69% were obtained on CT and PET-CT respectively with annotations from three expert radiation oncologists. While the data are different from the ones used in this paper, it provides an insight into the difficulty of the task and suggests that an automatic segmentation could learn a robust strategy from multiple annotators.

It is noteworthy to mention limitations of the dataset (one example illustrated in Figure 4 (b)) with annotations made for most cases on a planning CT then registered to the FDG-PET and CT volumes. In light of these results and limitations, this work sets the path to future larger studies with data curation of the present dataset, incorporation of other datasets from various centers that we plan to develop in a near future and comparison of various recent architectures and fusion strategies. An automatic pipeline to detect the oropharyngeal region in a full body scan will also be developed to bypass the limitation of the proposed approach that centers input volumes around the tumors.

In future work, we will also analyze the performance of tumors and metastatic nodes separately and evaluate the usability of the automatic segmentation algorithms for radiomics studies, starting with a comparison of the features extracted from manually annotated and automatically segmented ROIs. The features extracted from automatic segmentations will then be evaluated in standard radiomics studies in an attempt to uncover and predict disease characteristics. Finally, an extensive comparison of segmentation algorithms on these data will be carried out in the context of the HECKTOR challenge[2] at MICCAI 2020.

## Acknowledgments

This work was partially supported by the Swiss National Science Foundation (SNSF, grant 205320_179069) and the Swiss Personalized Health Network (SPHN via the IMAGINE and QA4IQI projects).

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
