# OpenReview forum: "Automatic Segmentation of Head and Neck Tumors and Nodal Metastases in PET-CT scans"
_MIDL.io/2020/Conference — MIDL 2020_

### Official Review · AnonReviewer3 · 2020-03-10
**Not well validated, comparisons are missing, some unjustified claims make the paper difficult to read at certain parts**

**Rating:** 2
**Confidence:** 5

**Summary:**

Authors presented a method for segmentation of head neck tumors and nodal metastases from PET-CT images. The basic idea is to use 2D and 3D V-Net, experiments were done on 202 patients' scans, and authors show there is an increase if two modalities are used together when segmenting. Interestingly, authors' 2D approach was slight better than 3D.

**Strengths:**

For radiomics, prediction of tumor growth and for several other clinical imaging perspective, the co-segmentation (or maybe better we call "joint segmentation") problem is important. Authors used 2D and 3D neural networks, with some fusion strategies to improve segmentations. One of the strengths of the paper is to have large number of patients evaluated. Using both 2D and 3D networks comparatively is also an application wise incremental addition to the paper.

**Weaknesses:**

The paper describes itself as a well-validated application; therefore, I will only briefly mention here that techniques that authors are using are not new.


--- having ground truths only from CT does not seem a fully feasible approach, it forces the system to learn tumor regions only when boundaries from PET and CT are very close to each other.
--- DSC is good, but not enough for a complete comparison of segmentation (and evaluation). A shape mismatch based metric is necessary too (or completely switch into FP and TP volume fractions).
--- The literature is not complete, several more recent co-segmentation (and joint segmentation works) are not cited, or authors are not aware of such works. For the completeness of the article, and being fair, those works, at least the state of the art methodologies should be mentioned and compared. For instance, C.Lian et al IEEE TIP 2018, Bagci et al MedIA 2013,  Guo et al IEEE 2019 TRPMS, etc....)



**Justification Of Rating:**

The paper is in the category of well validated application but this "well validated" component is missing in this paper.
 Some of the primary reasons are the following (summary)
-- unjustified claims about fusion strategies
-- preparation of ground truths labeling does not correlate with results, or authors fail to explain why.
-- it is cross validation study, generalization ability is not known, independent set is not used.
-- comparison with state of the art deep nets (particularly in this topic, not like basic u-net or v-net),  and pre-deep networks papers are missing.
-- evaluations are based on DSC scores only, there is a need for shape scores as well (and full version of DSC).


**Paper Type:**

validation/application paper

**Questions To Address In The Rebuttal:**

--- if all the data are delineated on CT of the PET/CT or other CT scans, it is hard to understand how PET-boundaries are better than CT at the end? In some other co-segmentation studies, experts seem to segment tumors using fused pet/ct to generate ground truths. It makes sense because metabolic and gross tumor regions can be seen better when combined. Same idea should apply for the auto-method for co-segmentation, complementary information can be obtained when combining multimodality in the algorithm. What I understand from the technique that authors use, tumors boundaries in pet and ct should sufficiently be close together so that neural networks can give good results, otherwise pet segmentation cannot beat really CT segmentation (because ground truths in authors' case come from CT). This raises a concern about the generalization properties of the network, because the network learn only CT segmentation, and since PET is a contrast image, it gives better boundary in the segmentation and results look better. This can be done with any settings, why neural networks then?

--- comparisons with pre-deep learning papers (like in Song's IPMI or Bagci's MedIA papers) as well as deep learning based co-segmentation papers can identify the position of this paper better, but authors only used V-Net and U-Net for their own implementation and nothing from the literature really. Therefore, it is hard to guess if deep network has some really superiority those that of graph based methods, and if V-Net and 2D V Net (I assume it is U-Net) are better than other specifically designed (for pet-ct cosegmentation) architectures or not. The results seem quite highly accurate in such papers (for different diseases but similar modalities such as PET/CT, and PET/MRI*).

--- Nyul 2000's standardization is applied to MRI, with known landmarks on the bimodal histograms. How do authors use the same algorithm to apply into PET and CT for standardization ? (authors used the terminology "normalization" but not sure if that is a correct terminology because standardization is non-linear while normalization is linear).

--- not well grounded why late fusion works better? when images look like similar (in authors' case, boundaries are close to each other), which means network may utilize such information better in early fusion, therefore not clear why it works better in late fusion.

--- can authors provide TP and FP volume fractions to better identify comparisons of the methods in tables and figures? DSC itself is not a completely good metric to understand evaluation of a segmentation, it should accompany with a shape mismatch term (like Hausdorff Distance) or complete terms in F-score (TP, FP,...).

--- in many parts, it is confusing what 2D V-Net, authors may clarify if they meant "U-Net" by 2D V-Net ?

--- is there any explanation why 2D V-Net (U-net?) is giving slightly better results than 3D V-Net ? authors mentioned this observation but no justification yet. Can authors look at the results from the size of the tumor and see in what conditions one network is doing better? maybe only in small tumors? maybe in large tumors?

--- alpha=0.6, when DSC is maximum, it means p-pred is more affected from CT, but results again show PET only segmentation is better than CT only segmentation...not justified why this is the case (if z axes resolution is the only reason, then it should have been the same for all cases, it only happens after late fusion, not early fusion or other cases).

---authors used cross validation, it is a feasible strategy but for well validated application studies there is a strong need to have completely blind data set (or called independent test set, in other terms) to test the generalization ability of the method. Please see also editorial comments By David Bluemke (Editor in Chief, Radiology) and others, 2020.


**Special Issue:**

no

---

> ### Author Response · Authors · 2020-03-27
> **Answer to review**
>
> Thank you for this review, important points were to clarify and results and comments were added to the document.
>
> Regarding the evaluation, Hausdorf distance is not defined for multiple regions and F1-score is the DSC score. We added precision and recall to complement the DSC results (Table 1 and Section 3.1). We also included per-center performance and will add results for small and large tumor volumes as suggested. We agree that there might be distinct subpopulations of tumors. We are currently computing the performance for tumors below 2cm and above 4cm of diameter, which correspond to the T1 and T3 TNM stages [2]. These results will be added in the coming days.
>
> [2] Huang, Shao Hui, and Brian O’Sullivan. "Overview of the 8th edition TNM classification for head and neck cancer." Current treatment options in oncology 18.7 (2017): 40.
>
>
> Regarding the literature, we agree that the proposed papers are related, yet we want to mainly focus on PET-CT and H&N tumor segmentation. We added a review paper (Zhou et. al 2019) of multi-modal medical image segmentation, including Lian et al IEEE TIP 2018, Guo et al IEEE 2019 TRPMS.
> Regarding the comparison with other methods, we developed a baseline method with standard deep learning architectures. This is an important step for larger studies that we will conduct, with curated and larger datasets, comparing various methods.
>
> For your comments on data annotation, the annotators have additional information for the CT annotation (clinical information and implicitly PET information, even when not viewer in fused). Given this context, the metabolic PET information is highly informative for the automatic annotation.
>
>
> For the standardization, the landmarks are calculated from the training PET and CT histograms for each split. It is not optimal and this method is better suited for MRI data but it improves performance also on the PET-CT data, as we also have bimodal distributions. Cutting off the tails and standardizing the data is a good practice also for these modalities. We will investigate this question more thoroughly in future work. We have renamed it to standardization as it is indeed the correct term.
>
> Regarding the “2D-VNet”: For fair comparison, we use a U-Net with a similar architecture as the V-Net (but 2D inputs). That is why we refer to it as 2D V-Net. We clarified this point in Section 2.3.
>
> Regarding the comment on alpha=0.6, the DSC is actually best for alpha = 0.4 as reported in the Figure and in the text. We changed the DSC scores in Fig 2 to percentages to be consistent with the rest of the paper and to avoid this possible confusion.
>
> Finally, we agree with the comment on the cross-validation. This work is only a baseline that is important for future comparison and further studies on these data. Since we do not optimize any hyper-parameter, we directly report the cross-validation results. We are currently developing an additional test set for further studies.

---

> > ### Comment · AnonReviewer3 · 2020-03-31
> > **partially responded rebuttal**
> >
> > I want to thank the authors to take time and responded some of my critiques.
> > Since there is limited time to do many experiments (and which I am not asking, but positioning the paper requires really some of the experiments to be done, hopefully in next version of the paper), and since authors clarified some issues related to the paper, and promising to add new results, I want to change my decision to weak accept. Please pay extra attention to include all relevant works. It is important to cite and give enough credits to all scientists who work in this field before and now. This way you will be improving visibility of this conference, and scientists in this field as well, nothing to mention the ethical part of it.
> >
> > --one concern: in PET images, I think SPIE 2020 articles now include some methods for standardization of PET images. I would be cautious when I use the MRI terminology in PET images.

---

### Official Review · AnonReviewer1 · 2020-03-12
**Review of paper: automatic segmentation of head and neck tumors and nodal metastases in pet-ct scans**

**Rating:** 3
**Confidence:** 4
**Recommendation:** Poster

**Summary:**

The authors investigate a way to automatically segment oropharynx tumors in the head and neck region to better identify the patients with a worse prognosis. On 202 cases with oropharynx tumors from four centers, a V-Net for segmentation was trained in 2D and 3D. They compare the segmentation results of CT, PET and CT + PET.

**Strengths:**

To prove that the combination of CT and PET-CT scans can improve patient care for patients with head and neck tumors, a large validation study is needed. For this clinical validation, a large number of cases are needed with delineated tumors, which is a very time-consuming process when done manually.  The study, therefore, investigated looked for a way to automatically delineate tumor outlines with data from four centers.

**Weaknesses:**

The paper is nicely written although there are some details that need to be addressed:
- For the dataset, the authors explain the cross-validation split. However, they don't mention the split between training & validation.
-  Could the authors explain how well the manual annotations overlap when a separate CT scan for treatment planning is used.
- The authors give averaged DSC for all centers. Could the authors also provide non-averaged DSC to see if the results are equal for all centers or there is one outlier.
- In figure 3 there is no visual example of the 'late fusion' method.
- The authors mention another paper focusing on head and neck tumors, if comparable, could they provide some comparison to their method and scores.
- In the discussion, it would be nice to see how the authors think the DSC could be improved.

**Justification Of Rating:**

The authors try to tackle a problem that hasn't been explored a lot with deep learning. They use an existing model with an open-source dataset to train and validate their results. The obtained scores leave some room for improvement but in the study is well setup.


**Paper Type:**

validation/application paper

**Special Issue:**

no

---

> ### Author Response · Authors · 2020-03-27
> **Answer to review**
>
> Thank you for the valuable reviews and suggestions.
>
> Regarding the cross-validation, we do not optimize the hyper-parameters (e.g. number of iterations, learning rate, etc.) of the networks thus directly report the results of the cross-validation. This paper is designed as a simple baseline for this task on the publicly available dataset. We are currently developing a separate test set for further studies.
>
> Regarding the overlap of manual annotations, this is a relevant question. Unfortunately, there is only one contour per case and we do not have this information. We are currently performing an inter-annotator agreement together with a data curation for future research built on top of this baseline paper and will investigate this point in the future.
>
> Following your suggestion, we now provide the per-center DSC scores and comment on it in Section 3.1. The least represented center has the lowest average DSC score. The annotations of the 19 cases of this center particularly require a quality control and more training data similar to those of this center may also be required for better generalization to these cases.
>
> For Figure 3, the late fusion is an average of the PET and CT and we think it is less informative than the early fusion.
>
> The dataset used in Moe et. al 2019 is different from our dataset and not publicly available. The results are not comparable.
>
> Finally, we foresee an improvement of the performance with a data curation, larger cohorts, and the use of recent architectures with various fusion methods. We added a comment on this in the discussion as suggested.

---

### Official Review · AnonReviewer2 · 2020-03-13
**An automated deep learning based segmentation approach for head and  neck tumor and nodal metastases in PET-CT scans.**

**Rating:** 3
**Confidence:** 4
**Recommendation:** Poster

**Summary:**

This work proposes a deep learning approach for automated segmentation of tumors and nodal metastases in head and neck scans. Two approaches are proposed that utilize both CT and PET (early and late fusion) and these are compared to using only CT or PET. Furthermore, 2D and 3D methods are compared. The results demonstrate the fusion approaches outperform single modality. Suprisingly, the 2D method outperforms the 3D method.

**Strengths:**

The comparison of early fusion vs late fusion vs single modality is interesting and provides valuable insight into what features are important for this task. The comparison to 2D vs 3D is also interesting, however, the results are surprising and more insight to this would be beneficial.

**Weaknesses:**

The main weakness is the lack of novelty of the proposed method. Both fusion methods have been proposed before. It would be interesting to see a more advanced fusion approach - using input channels for early fusion and average masks for late fusion are quite simple solutions (which is not necessary a bad thing, it is possible these are the best solutions).



**Detailed Comments:**

Acronyms should be defined upon first use (abstract: FDG-PET and CT)

**Justification Of Rating:**

The evaluation and comparison of different methods (including different fusion techniques vs single modality and 2D vs 3d) is thorough and interesting. However, the novelty of the proposed method is very limited. Therefore, I recommend this work to be presented as a poster.

**Paper Type:**

validation/application paper

**Questions To Address In The Rebuttal:**

The method is dependent on pre-processing that requires calculating a bounding box around the ground truth segmentation. Therefore, it is unclear how the method will perform on new cases where ground truth
 segmentation is not available for cropping. Will this require an additional automated step for localization? Or will a manual step be required? Authors should provide insight to this and discuss sensitivity to the initial bounding box step.

More evaluation metrics should be included to better understand performance of the method. Surface distance between ground truth and prediction should be included as a metric of distance between the boundaries. Furthermore, true positive and false positive metrics should be included (such as area under the curve) to better understand if there is any bias in each method for over- or under-segmentation. With this application in mind, such metrics could also be used to optimize the alpha value (is it better to miss tumor or over-segment tumor?). This should be discussed in paper.

Any insight to why 2D performs better than 3D? It would be interesting to include 3D renderings comparing 2D vs 3D. While
 2D may slightly outperform 3D quantitatively in terms of Dice coefficient, 3D may have the advantage of producing more consistent results in 3D.

**Special Issue:**

no

---

> ### Author Response · Authors · 2020-03-27
> **Answer to review**
>
> Thank you for your comments and suggestions
>
> This paper serves as a baseline with the current publicly available data. We are currently running a data curation for further studies, as well as developing an automatic cropping method based on brain detection. However, the bounding box is large and the tumors are easily contained in it. Therefore, we expect the algorithm to generalize well to an automatic or semi-automatic region bounding box.
>
> The question of under or over-segmentation depends a lot on the application and location. For example in radiotherapy (but also radiomics), the regions are generally larger than the tumor, but some structures must be avoided, e.g. salivary glands [1].
> In radiomics, the border of the tumor can be informative as well as the volume of interest. In this context, it can be considered better to over-segment. It also depends a lot on the feature we consider. In these data, the contours were annotated for radiotherapy and the contours are already large, over-segmentation may be considered worse than under-segmentation. We have now added an evaluation of precision and recall (Table 1), showing that 3D methods tend to over-segment more than the 2D ones and comment on this point (Section 3.1).
>
> Regarding the inconsistency of 2D segmentations, it is the case for CT segmentations. PET and PET-CT segmentations, however, rely a lot on PET intensity which is consistent across slices resulting in rather consistent 3D volumes.
>
> [1] Grundmann, O., G. C. Mitchell, and K. H. Limesand. "Sensitivity of salivary glands to radiation: from animal models to therapies." Journal of dental research 88.10 (2009): 894-903.

---

### Official Review · AnonReviewer4 · 2020-03-18
**Interesting application with reasonable validation**

**Rating:** 3
**Confidence:** 4
**Recommendation:** Poster

**Summary:**

The work was an interesting use of V-Net on a challenging problem. PET-CT is a challenging machine learning task because it has two large resolution images with different anisotropic resolution and so requires quite a few decisions and trade-offs. The paper did a good job of re-using established, validated tools and making the code and data available to allow for reproduction of the results. The validation aspect was generally good but was missing some of the thoroughness expected in medical papers. In particular, the lack of any kind of baseline comparison or lesion detection level metrics made it difficult to appreciate the degree of success the method had.

**Strengths:**

The strengths were clearly described pre-processing and model selection steps. The methods section was sufficiently detailed to recreate their steps independently. The approach comparing 2D and 3D and different fusion methods was also very thorough compared to similar works. The figures were well done and easy to read and interpret.

**Weaknesses:**

There were a few weaknesses to the paper. Principally very little context was given to their model performance. A DSC of 0.61 could be fantastic or terrible but without knowing the general range of expert human performance and a simple threshold and/or classical computer vision on PET it is difficult to assess how well the model performed. Furthermore from a clinical standpoint, the results are not presented in a physiologically meaningful manner. Did it miss 40% of the lesions? Did it estimate the lesion volume 40% lower than it was? Did it find all tumors but miss all metastases? Without knowing these specifics it would be very difficult to show that such a model would offer any value at all to a clinician.

The use of a paired T-test for comparing fusion approaches seemed dubious at best and I would leave it out.

The decision to use isotropic sizes despite the data's anisotropic acquisition was potentially not justified and probably hindered 3D performance.

The use of publically available data was good, but means there is little understanding of the errors and problems with the ground-truth labels. The use of multiple physicians to provide rough estimates of interreader variability would have massively strengthened this work.

**Justification Of Rating:**

The paper was well-written and easy to understand and follow. The steps were well documented and the results would be of some interest to others in the field working with similar 3D and/or multiple-contrast fusion problems. The clinical relevance was unclear and the impact without more robust comparisons is hard to assess.

**Paper Type:**

validation/application paper

**Questions To Address In The Rebuttal:**

Better identification of how the model's performance can be interpreted across different classes of lesions and different people and a clearer message on which cases worked well and where the model failed.



**Special Issue:**

no

---

> ### Author Response · Authors · 2020-03-27
> **Answers to comments and modifications**
>
>
> Thank you for these relevant comments and suggestions.
>
> For a better analysis of the performance, we have now added the precision and recall measures to the DSC (Table 1) as well as per-center performance and further discussion (Section 3.1). We will also include an analysis of results on small and large tumors in the coming days.
>
> Regarding the statistical test, we think it is appropriate. The central limit theorem holds and it shows that the DSC comparisons are statistically significant. We removed one comparison and reformulated. We hope it is now easier to read.
>
> Resampling is used to unify the spacing between cases. It is a rather standard approach to use isotropic resampling but other resampling approaches could indeed be used in the future.
>
> Finally, regarding the data, we are currently working on cleaning the data with a quality control for further studies and developing an inter-annotator agreement as well as including more cases. This paper serves as an important baseline with the current data and using basic methods. We agree that human level performance is unknown at this point. However, it is worth noting that even in the case where the approach is reasonably less accurate than the top human observers, the fully automatic approach is still very valuable to allow reproducible radiomics studies with very large patient cohorts.

---

### Meta-Review · Area_Chair1 · 2020-04-08
**MetaReview of Paper205 by AreaChair1**

**Rating:** 4
**Recommendation For Accepted Papers:** Poster

**Metareview:**

We have four reviewers voting for an accept (one of them updating a 'weak accept' after the authors have responded to her/his critique).

I would agree and recommend an 'accept' as well.

**Paper Type:**

both

**Special Issue:**

no

---

### Decision · Program_Chairs · 2020-04-11

Accept